# HPV and Cervical Cancer: Molecular and Immunological Aspects, Epidemiology and Effect of Vaccination in Latin American Women

**DOI:** 10.3390/v16030327

**Published:** 2024-02-21

**Authors:** Christian David Hernández-Silva, Adrián Ramírez de Arellano, Ana Laura Pereira-Suárez, Inocencia Guadalupe Ramírez-López

**Affiliations:** 1Departamento de Microbiología y Patología, Centro Universitario de Ciencias de la Salud, Universidad de Guadalajara, Guadalajara 44340, Jalisco, Mexico; christian.hernandez@academicos.udg.mx (C.D.H.-S.); ana.pereira@academicos.udg.mx (A.L.P.-S.); 2Instituto de Investigación en Ciencias Biomédicas, Centro Universitario de Ciencias de la Salud, Universidad de Guadalajara, Guadalajara 44340, Jalisco, Mexico; adrian.ramirez@academicos.udg.mx; 3Departamento de Ciencias de La Salud, CUValles, Universidad de Guadalajara, Guadalajara-Ameca Rd Km. 45.5, Ameca 46600, Jalisco, Mexico

**Keywords:** HPV, epidemiology, cervical cancer, immunological aspects, vaccines, Latin America

## Abstract

Cervical cancer is primarily caused by Human Papillomavirus (HPV) infection and remains a significant public health concern, particularly in Latin American regions. This comprehensive narrative review addresses the relationship between Human Papillomavirus (HPV) and cervical cancer, focusing on Latin American women. It explores molecular and immunological aspects of HPV infection, its role in cervical cancer development, and the epidemiology in this region, highlighting the prevalence and diversity of HPV genotypes. The impact of vaccination initiatives on cervical cancer rates in Latin America is critically evaluated. The advent of HPV vaccines has presented a significant tool in combating the burden of this malignancy, with notable successes observed in various countries, the latter due to their impact on immune responses. The review synthesizes current knowledge, emphasizes the importance of continued research and strategies for cervical cancer prevention, and underscores the need for ongoing efforts in this field.

## 1. Introduction

Human Papillomavirus (HPV) represents a significant global health challenge, particularly in its association with cervical cancer (CC), a leading cause of women’s mortality worldwide [1]. In 2020, approximately 604,000 new cases and 342,000 deaths were attributed to this disease [2]. Understanding how HPV genes interact with the immune system sheds light on the mechanisms behind CC development.

HPV transmission occurs through skin-to-skin and skin-to-mucous membrane contact. Among the main routes are horizontal transmission (which includes fomites and skin, but not through sexual contact), autoinoculation, vertical transmission (from mother to newborn), and sexual transmission, the latter being the most documented and well-known [3].

The presence of HPV infection in one anatomical site represents a great risk for HPV infection in another anatomical site. The highest risk factor for cervical HPV infection is the presence of anal HPV infection and, conversely, the presence of cervical HPV infection is a major risk factor for anal HPV infection [4,5].

The HPV lifecycle begins with the virus interacting with host cells, particularly keratinocytes in the body’s layered tissues. HPV capitalizes on minor tissue injuries, such as those during sexual activity, binding to host cells and starting its replication and transcription process [6]. The early expression region (E) of HPV encodes various proteins—E1, E2, E1ˆE4, E5, E6, E7, and E8ˆE2—each playing a unique role in the virus’ lifecycle. Particularly, E6 and E7 oncoproteins play a central role in causing changes in host cells, leading to cancer [7].

Understanding how HPV evades the immune system is crucial. While the immune system clears most HPV infections, about 10% persist, leading to lesions that may turn cancerous. HPV uses various tactics, such as altering immune cell recognition and suppressing immune responses, to avoid detection [7,8,9,10]).

Moreover, HPV employs multifaceted strategies to evade the body’s immune responses, including manipulating immune sensors and altering cytokines involved in inflammation and cancer progression. This complex interplay between HPV genes and the immune system significantly influences CC development and potential treatments [11].

Vaccination stands as a powerful tool in the global fight against Human Papillomavirus (HPV) infections, significantly reducing the risk of HPV-related cancers. Despite this, the prevalence of diseases linked to HPV infections, notably CC, persists as a major health concern [12]. In fact, CC ranks as the fourth most common cancer in women globally [1]. It is also important to mention that a significant risk of extra-cervical tumors, including those in the anus, vagina, vulva, oropharynx, as well as the lungs and bladder has been observed in women infected with HPV, which supports the need for early diagnosis and screening for these cancers in higher-risk populations [13].

To address this challenge, various countries have implemented vaccination campaigns targeting HPV-related diseases. However, achieving comprehensive success in these efforts has proven to be more intricate than anticipated. Before delving into the current situation in Latin America (LA) and the obstacles to achieving vaccination goals, understanding the key prophylactic vaccines currently in use is essential [12].

However, despite the widespread use of vaccines such as Cervarix^®^, Gardasil^®^, and Gardasil 9^®^, challenges remain in achieving optimal vaccination rates [14,15,16]. Concerns over safety, misconceptions about its effects on sexual behavior, and limited awareness hinder vaccination efforts in Latin American countries. Overcoming these barriers necessitates targeted educational campaigns, healthcare provider training, and community engagement initiatives [17,18,19].

To combat these challenges effectively, leveraging strategies involving multi-sectoral partnerships, evidence-driven decision-making, and community involvement is crucial. Addressing vaccination disparities, improving monitoring systems, and enhancing awareness are paramount to achieving comprehensive CC prevention in LA [20].

This review explores the intricate relationships between HPV genes, their role in viral replication, immune evasion strategies employed by the virus, and their profound impact on CC in Latin American women. While this review adopts a narrative approach rather than a systematic one, we meticulously curated articles within a defined timeframe. Specifically, for the examination of vaccination in Latin American countries, emphasis was placed on recent publications (2016 to present). The scope of the review deliberately excludes information pertaining to countries beyond LA. This deeper understanding offers hope for targeted treatments and interventions to reduce the burden of HPV-related diseases, particularly CC.

## 2. Role of the HPV Genes on the Viral Replication and Cervical Carcinogenesis

According to the International Agency for Research on Cancer (IARC), among the vast range of over 200 known HPV genotypes, some are classified as high-risk HPV (HR-HPV), HPV-16 and HPV-18 being the most common, which are strongly associated with around 70% of CC cases. These genotypes represent approximately 70% of the cases, but the list includes at least ten others, such as 31, 33, 35, 39, 45, 51, 52, 56, 58, and 59 [6,7,21,22].

HPV is a small virus (with a 50–60 nm diameter) that is non-enveloped and icosahedral, and whose genome consists of a single circular double-stranded DNA molecule of approximately 8000 base pairs with eight open reading frames and containing three genic regions: one long control region (LCR) or upstream regulatory region (URR), an early expression region (E), and a late expression region (L) [23].

HPV infects keratinocytes in the basal lamina of stratified mucoepithelial tissues. Virions access the basal cells through microlesions in the epithelium, mainly occurring during sexual intercourse, allowing virions to bind to heparan sulfate proteoglycans (HSPG) by interacting with the viral protein L1. After the interaction with L1, the L2 protein is exposed, and it participates in endocytosis and intracellular trafficking, via the Golgi apparatus, to finally enter the cell nucleus and initiate the viral gene replication and transcription [6,7,21,24].

The URR contains the origin of replication (Ori), as well as binding sites for E1, E2, and cellular transcription factors (including NFI, Oct-1, AP-1, TEF-1, and SP1), so it is responsible for controlling the virus replication and gene transcription [21,25]. The E region encodes for non-structural proteins: E1, E2, E1ˆE4, E5, E6, E7, and E8ˆE2 [7].

E1 has helicase activity, and it acts through the recognition of the Ori sequence, with E2 being an accessory protein for E1, which is why it is responsible for regulating the replication of the virus by recognizing the sequence of origin; furthermore, E2 functions as an activating or repressing transcription factor [21].

E1ˆE4 is expressed late during the infection to facilitate the virus’ exit and transmission by altering the cytokeratin network of host cells. However, it has also been involved in the cell cycle arrest in the G2/M phase, in the amplification of viral replication, sustained activation of the mitogen-activated protein kinase (MAPK) signaling pathway, and stabilization of the E2 protein [11,21,23].

E5, E6, and E7 are the three oncoproteins encoded by HPV that promote keratinocyte transformation and immortalization [21,23].

The function of the E5 oncoprotein has not been fully characterized yet; however, it has been observed to play an essential role in stimulating keratinocyte proliferation by activating the MAPK signaling pathway by interacting with epidermal growth factor receptors and platelet-derived growth factor receptors (EGFR and PDGFR) and the phosphorylation of crucial transcription factors, such as c-fos, c-myc, Ets1, Ets2, Elk-1, and c-jun [9,26]. E5 has also been related to the immune response evasion towards the virus, the inhibition of apoptosis, and increased cell motility. Despite all the mentioned effects, it is essential to highlight that E5 does not directly contribute to the carcinogenic potential of HPV [26,27,28].

The viral oncoproteins E6 and E7 cooperate directly for the oncogenic transformation of HR-HPV-infected cells [11]. These oncoproteins are expressed at low levels early; however, in differentiated epithelial cells, they stimulate the cell cycle to overcome checkpoints and promote their reproduction by targeting tumor suppressor proteins [6,21]. The E6 oncoprotein promotes the ubiquitination of the tumor suppressor p53 and its degradation via proteasome and avoids cell death. Furthermore, E6 promotes the expression of human telomerase reverse transcriptase (hTERT), which is involved in immortalizing the infected keratinocyte. On the other hand, the E7 oncoprotein induces the degradation via proteasome of the retinoblastoma protein (pRb) and deregulation of the E2F factor, cyclin A/CDK2, and cyclin E/Cdk2, causing uncontrolled cell proliferation [9,11,29].

The activity of E8ˆE2 is not fully described. However, it has been found to be an inhibitor of the viral gene expression and HPV replication, so its primary function is to limit the number of virions produced, thus determining latent or productive infection [30].

Finally, the L region codes for two structural proteins of the capsid, L1 (the majority protein) and L2. These proteins comprise a capsid composed of 72 pentameric capsomers [21,23].

## 3. Immune Response and Immune Evasion in Cervical Cancer

Approximately 10% of women with an HPV infection develop a persistent infection, which can potentially lead to the formation of low-grade intraepithelial lesions (LSIL or CIN 1, Cervical Intraepithelial Neoplasia grade 1) or high-grade intraepithelial lesions (HSIL or Cervical Intraepithelial Neoplasia grade 2 and 3). However, it is important to note that only a subset of these women with persistent infections and high-grade lesions will eventually develop cervical cancer. While the lifetime risk of acquiring an HPV infection is estimated to be around 70–80%, the actual incidence of cervical cancer without screening is much lower, affecting only about 2–3% of all women. This indicates that the immune system is capable of eliminating HPV in the majority of cases, preventing the progression to cervical cancer in approximately 90% of infections [7,8,9,10,31].

Among the main factors promoting this immune evasion is the virus replication cycle separately from immune cells. HPV is maintained episomally in basal cells with minimal expression of viral proteins, induced by the repression of early promoters by the viral protein E2. Similarly, HPV infection does not produce keratinocyte lysis, so it does not induce inflammation, evading the cellular and humoral immune responses [6,27,29].

E5 oncoprotein from HPV16 downregulates the A and B alleles of the human leukocyte antigens (HLA-A and B) from the major histocompatibility complex (MHC) class I by interaction with the heavy chain and its retention in the Golgi apparatus, thus preventing the presentation of viral antigens to CD8+ T lymphocytes. Meanwhile, the E6 oncoprotein inhibits the expression of MHC-I genes [32,33,34].

It has been reported that the inhibition of MHC-I by HPV oncoproteins is reversed by stimulation with IFN; however, it was also found that E6 and E7 oncoproteins inhibit the production of IFN-I—mainly IFN-β and IFN-κ [27,35,36]—while E1 suppresses IFN-β and IFN-λ [37]. This process, in turn, generates an immunosuppressive environment in which cytokines such as IL-10 and transforming growth factor β (TGF-β) participate. These cytokines are produced mainly by regulatory T cells (Treg), inhibiting the antigen presentation function [8,10].

The reduction in HLA-A/B is a stimulatory signal for cell death by NK cells. The C and E alleles of HLA inhibit the activation of NK cells, and it has been found that the E5 oncoprotein does not reduce the expression of MHC-I, which is why it is considered a mechanism for the evasion of missing self in HPV-infected cells [29,33,38,39].

The stimulation of pattern recognition receptors (PRRs) by molecular patterns associated with pathogens, and damage (PAMPs and DAMPs) by HPV constitutes the first line of defense against infection, among them, Toll-like receptors (TLRs) and cytosolic DNA sensors [7].

The activation of TLRs by HPV induces the production of proinflammatory cytokines that help defend against the virus. However, HPV16 causes a decrease in the expression of TLR2 and TLR7 and increases the expression of TLR4 [40]. Although TLR4 has been involved in the clearance of infection by the induction of inflammasomes that interfere with the integration of the viral DNA into the DNA of the host cell, as well as the isotype change of immunoglobulins against HPV, other studies have found that TLR4 is downregulated by HPV, decreasing the expression of proinflammatory cytokines such as IL-6 and IL-8 [41,42]. Furthermore, the E7 oncoprotein of HPV16 causes gene silencing of TLR9, which participates in the detection of double-stranded DNA, which cooperates in the inhibition of IFN-I and also suppresses the activation of NF-κB, also preventing the production of proinflammatory cytokines [29].

One of the viral DNA detection pathways is cGAS-STING; however, the E7 oncoprotein of HPV18 inhibits this signaling and blocks the expression of proinflammatory cytokines and IFN-I [29,43].

The E6 and E7 oncoproteins cooperate to release proinflammatory cytokines such as IL-1β, IL-6, and IL-18. Furthermore, these oncoproteins decrease the expression of IL-2 and IL-17. IL-2 is essential for the development of the cellular adaptive immune response [44]. For its part, IL-17 is a proinflammatory cytokine that can help clear the infection; however, it has also been reported that, in persistent HPV infections, it can promote protumorigenic properties [7,45,46]. IL-6 and IL-18 can promote tumor progression of cervical lesions in the context of chronic inflammation [7,11].

The viral protein E2 and oncoproteins E6 and E7 reduce the migration, maturation, and differentiation of innate immune cells such as dendritic cells and monocytes/macrophages, for example, by the overproduction of prostaglandin E2 [7,8].

Another level at which immune regulation by HPV and CC acts is the increase in the expression of the immune checkpoints CTLA-4 and PD-L1 on the membrane of antigen-presenting cells and infected cells, the effect of which is to inhibit cytotoxic T lymphocyte activation and Treg induction [10].

## 4. Hormonal Influences on Cervical Cancer Risk

Although persistent HPV infection is necessary for the development of CC, it is not sufficient, and a series of co-factors increase the risk of carcinogenesis, including age, ethnic origin, poor hygiene, early onset of sexual life, number of pregnancies and births, coinfections (*Chlamydia trachomatis*, *Trichomonas vaginalis*, and *Herpes Simplex Virus* type 2), smoking, exposure to diethylstilbestrol, and hormonal factors [28,47,48]. Among the main hormonal factors that are related to the development of CC are the prolonged use of oral contraceptives as well as exposure to hormones, including estrogens, prolactin, and progesterone. However, their role is still not completely clear [49,50,51].

Oral contraceptives prevent pregnancy by blocking the maturation of the ovarian follicle, thus preventing ovulation. The most common birth control pill is made up of a mixture of estrogens and progestins, a synthetic form of progesterone; however, some contraceptives contain only progestins [52].

In 2002, an IARC case-control study showed a strong relationship between the duration of oral contraceptive use and a risk of cervical cancer in women with cervical HPV infection [53]. Similarly, results from the European Prospective Investigation into Cancer and Nutrition (EPIC) cohort study showed that prolonged use of oral contraceptives significantly increased the development of grade 3 cervical intraepithelial neoplasia, carcinoma in situ, and invasive carcinoma. This study also showed that 17β-estradiol (E2) levels could be involved in the development of invasive cervical carcinoma [54]. Other studies found that this risk occurs when oral contraceptives are used for more than five years and in HPV-positive patients [55].

Some mechanisms have been proposed to promote the progression from precancerous lesions to malignancy by integrating the viral genome into the DNA of the host cells and increasing the expression of the viral oncoproteins E6 and E7 [56].

Progesterone increased the expression of the HPV16 E6 and E7 genes in the CaSki CC cell line and high levels of progesterone receptors in neoplastic lesions [47].

Estrogens act through two types of receptors; the canonical ones, estrogen receptor α and estrogen receptor β (ERα and ERβ), and a non-canonical receptor, the G protein-coupled estrogen receptor (GPER) [51]. In murine models, estrogen signaling is necessary for cervical carcinogenesis, mainly through the stimulation of ERα [57,58,59]. It was also found that estrogens cooperate with E5, E6, and E7 oncogenes and synergize with the reduction of p53 to induce said carcinogenesis [60,61,62]. It was subsequently observed that ERα signaling is vital in the development of CC not only for the HPV-infected cells, but that its activation is also essential in stromal cells [63].

Studies in murine models have also determined that selective ER antagonists and modulators were able to eliminate cancer cells and prevent the development of CC [64].

It has been found that ERα, ERβ, and GPER receptors are expressed in CC and its precursor lesions and that, in addition, their expression increases as the malignancy of the lesion progresses [51,65]. It was also reported that the E6 and E7 oncogenes of HPV16 and HPV18 increase the expression of ERα, PRLR, and GPER and modify their cellular localization [66]. Furthermore, E2 induces a dysfunction at the mitochondrial level that results in favoring the Warburg effect in CC cells [67].

In addition to studies on the importance of ERα, some have been carried out on the expression and activation of GPER. GPER activation induces cell cycle arrest [68]. Moreover, GPER stimulation reduces cell proliferation and induces apoptosis, necrosis, and senescence in CC cell lines [51]. It was recently shown that GPER silencing increases the protumor properties of CC cells, including migration, invasion, and epithelial-mesenchymal transition [69].

Prolactin (PRL) acts through a series of receptors (PRLRs), and its effects on gynecological cancers include migration, invasion, metastasis, inhibition of apoptosis, and chemoresistance [70]. Since then, several studies have contributed to the evidence of the participation of PRL in cervical carcinogenesis. Between 1970 and 2000, PRL was found in premalignant and malignant lesions of the cervix and the normal cervix, but its expression was lower in the latter [71]. Also, it was determined that PRL increases the proliferation of CC cell lines and cells from cervical carcinomas induced in mice [72,73,74,75]. Furthermore, it was observed that PRL is elevated in the advanced stages of the disease, that it decreases with surgical resection of the tumor, and that it is produced extrapituitarily by CC cells [76,77].

It was later shown that different PRLR isoforms are expressed, and PRL variants are produced in CC cell lines [78]. Additionally, PRLR expression in cervical tissue correlates with the stage of carcinogenesis [65,79].

The effects of recombinant PRL and a 60 kDa variant produced by CC cells have also been studied. The stimulation of CC cell lines only with recombinant PRL demonstrated a protective effect against apoptosis [80], while the 60 kDa variant inhibited apoptosis and induced the proliferation of the SiHa cell line [81].

In the context of hormonal cooperation, E2 and 60 kDa PRL modify the metabolism but do not affect the survival of cancer cells [65] and increase the expression of the E6 and E7 oncogenes of HPV16 and HPV18 [66].

Finally, E2 reduced cellular cytotoxicity in NKL cells, while PRL increased this effect. Furthermore, PRL decreased the expression of NKG2D in NKL cells and increased the expression of MICA/B, an effect that was reduced in combination with E2 in HeLa cells, while E2 and PRL decreased the levels of soluble MICA [82].

## 5. HPV and Cervical Cancer in Latin America

Despite all the advances that have been made in the early detection of HPV infection, CC remains one of the most common types of cancer in women. Specifically in the region of Latin America and the Caribbean, in 2020, GLOBOCAN reported 59,439 new cases of CC; in terms of the prevalence calculated over three years, 107,135 cases were found as well as an estimated 31,582 deaths, which leaves it in third place of incidence, the fourth most common type of cancer and at fourth place in mortality in the region [83].

Notwithstanding the trends in the region, in some countries such as Honduras, El Salvador, Nicaragua, Panama, Peru, and Paraguay, CC is the second most common type of cancer; in addition, in Bolivia, it stands out as the first. In addition, in Bolivia, CC has the first place in incidence, while in others, such as Mexico, Nicaragua, El Salvador, Honduras, Panama, Peru, Paraguay, and Venezuela, this type of cancer has second place. On the other hand, there are countries in which CC has the first place in mortality, such as Belize, Honduras, Nicaragua, Peru, Bolivia, and Paraguay [83].

Studies carried out in young (under 25 years), unvaccinated, and sexually active Latin American women reported a high prevalence of HPV (greater than 50%); for example, in Paraguay, the prevalence was 54.8% [84], in Brazil 54.6% [85], in Colombia it was 60.3% [86], and in Argentina it was 56.3% [87]. By contrast, studies carried out on women who attend routine checkups and are over 25 years old found that this percentage is lower, as is the case of the ESTAMPA subcohort study realized in Argentina, Colombia, Costa Rica, Honduras, Paraguay, and Uruguay when only 13% of samples were HPV-positive [88]. Something similar happened in a Cuban study in which a higher percentage of HPV-positive samples were observed in women under 25 years compared to the group of women over 25 [89]; while in Guatemala, similar findings were observed in a group of women under 30 [90].

The ESTAMPA subcohort found that HPV infection prevalence increased with cytology severity, indicating that the percentage of HPV-positive samples is higher in CC than CIN3, CIN2, and CIN1 [91]; they also found a phenomenon whereby HPV prevalence decreases with age, and HPV-positive samples are most commonly found in young women [91,92]. 

Although there is a general trend, it is essential to mention that there are differences in the reported prevalence within the same country; for example, a study in Mexico analyzed samples from 20 different states and found differences in the prevalence, highlighting Chihuahua and Nayarit as the states with the highest prevalence (28.74% and 28.33%, respectively). In contrast, Quintana Roo and Sinaloa showed a lower prevalence (21.17% and 21.54%, respectively) [93].

The HPV16 genotype is one of the most frequent in studies that evaluate distribution in young women before vaccination and, in those who attend routine check-ups, it is a disturbing fact because the leading indicator and predictor of the development of CIN3 is the persistence of infection with HPV16 [88]. On the other hand, different studies have shown that HPV16 is present in at least 50% of CC cases [88,94,95].

In studies conducted in Mexico, Argentina, Colombia, Costa Rica, Honduras, Paraguay, and Uruguay that evaluated the HPV distribution in cervical lesions, HPV16 stood out as the most prevalent genotype in all study groups [88,93,95]. In addition, the ESTAMPA study evidences an increase in HPV16 prevalence corresponding to histological grade and reveals that, in CIN3, the risk of infection with this genotype increases with age [88].

Concerning HPV genotype distribution, the prevalence of particular genotypes is unpredictable in each country and even in the different regions into which a nation is divided; for example, a study carried out in three areas from Cuba (Holguin, Havana, and Villa Clara) found differences in the number of coinfections, from which Holguin stands out for having a lower percentage of coinfections and a marked presence of HPV 89 compared to Havana and Villa Clara [89].

It seems that in epidemiologic studies, the nations are often subdivided for research purposes. In the case of HPV genotypes, they seem to have different distribution patterns even within the same area. For example, in a study in western Mexico, Molina et al. reported that, although HPV16 was present in first place in terms of frequency in almost all states that comprise the region, in Nayarit state, a frequent form of HPV was HPV56. In contrast, in Colima state, this HPV did not appear [95].

Although HPV genotype distribution in Latin America is diverse and not homogeneous between different countries, the research consulted shows that, broadly speaking, the most frequent genotypes, apart from HPV16, are 66, 59, 56, 52, 51, 42, 6, 44, and 51; for more details related to its prevalence in each country and the characteristics of the study in which it was found, see Table 1. In addition, although HPV18 does not figure as one of the most frequent in asymptomatic women, it has been seen that it is very present in CC samples [88,94,95].

The prevalence and distribution of HPV genotypes is a phenomenon that has been studied primarily in women. Still, if we look at the infectious cycle of HPV, it is more than evident that men also play a fundamental role in this story. Although this phenomenon is not studied in the same manner as in women—not enough—some studies show us some interesting data.

In terms of prevalence, HPV is highly present in men. In a study carried out on men with partners who presented cervical intraepithelial lesions, 41.9% of positive samples were found [98], while in another study carried out on young men under 25 years, the positive samples reached 51.8%; these last data are somewhat related to reports in women of the same age range in whom the percentage of positivity is higher too [85]. Similar data were reported in a study in Ecuador in which the percentage of positive men was approximately 63.5%, much higher than what was reported in women in the same study (39.5%) and, although they did not analyze the prevalence by age and sex, it can be seen that the percentage of positive samples is higher in the 18–30-year-old group [97].

Taking into account the natural history of HPV infection, the most logical thought would be that stable couples share the same HPV arsenal. However, a Colombia study carried out in heterosexual couples in which the woman presented cervical intraepithelial lesions reported that only 28% of couples showed concordance in at least 1 HPV, which is probably due to differences between men and women in the infection resolution time. Additionally, the co-infection was higher in men than women (64% and 30%, respectively); men with multiple infections reported having more than five sexual partners throughout their lives, which is a factor related to HPV prevalence [96].

Although the diverse behavior in the prevalence and frequency of the different HPV genotypes in LA is indisputable, the methodology used for both the collection and analysis of the samples could impact the results; for example, Martins et al. reported on a random subsampling of 223 HPV negative samples that they analyzed previously using HC2 or COBAS and subsequently sent for genotyping with the PapilloCheck test and, to their surprise, 13.9% of these initial negative samples were positive for some HPV genotypes. Additionally, 49 samples were initially HPV-positive and not in genotyping [92]. These results indicate that the diagnosis and prevalence reported in studies may not represent accurate data.

## 6. HPV Vaccines: Molecular Aspects of FDA Approved Vaccines

Vaccination is one of the most efficient strategies worldwide for reducing HPV-related cancers and infections. Unfortunately, the high incidence of diseases resulting from HPV infections remains a major concern, contributing to CC being the fourth most common cancer among women worldwide [1].

The implementation of vaccine campaigns has been an adequate strategy adopted by several countries attempting to reduce the incidence of HPV-related diseases. Nevertheless, this goal has not been completely achieved, and the explanation for it is more complex than it looks at first glimpse. In particular, the situation in Latin American countries is not optimal and this is a consequence of several factors; however, before deeply delving into the current situation in Latin America and the challenges to achieving that goal, it is convenient to describe the main prophylactic vaccines currently administered.

Cervarix^®^ (GSK), Gardasil^®^ (Merck), and Gardasil 9^®^ (Merck) are preventive vaccines directed to certain serotypes of HPV and they are designed as virus-like particles (VLP), meaning they resemble the external part of the HPV but do not have any infectious material in the inner side, making the vaccination safer [99]. The outer side of the VLP is comprised of an L1 viral protein produced by recombinant technology and it is recognized by the immune cells to generate antibodies able to protect vs. future infections [100,101,102].

Cervarix^®^ (GSK), approved by the FDA in 2009, is a bivalent vaccine targeting HPV 16 and 18, which are responsible for approximately 70% of CC [103]. Cervarix vaccination has been reported to offer a protection for up to 10 years by maintaining anti HPV 16 and 18 titers [14,104]. This vaccine has also demonstrated robust protection with significantly lower HPV prevalence in vaccinated women after four years, highlighting its potential in preventing HPV-associated oropharyngeal cancers [105].

Gardasil^®^ (Merck) is a quadrivalent vaccine used to prevent infections by HPV-6, 11, 16, and 18, which altogether represent the vast majority (main cause) of cutaneous lesions and cervical malignancies [15]. Additionally, this vaccine has proved to be useful in reducing oral cavity, penis, vulva, and anus HPV infections [106,107,108].

Besides the quadrivalent, there is another Gardasil^®^ vaccine (Gardasil 9^®^ by Merck) directed to nine HPV serotypes, including HPV-6, 11, 16, 18, 31, 33, 45, 53, and 58, covering approximately 90% of cervical malignancies [109].

Even though the most popular types of vaccines are the prophylactic ones, used to prevent infection by HPVs, there are other kinds of vaccines called “therapeutic vaccines”, useful in treating patients who are already infected and present some symptoms caused by HPVs. Among the most relevant ones are the following:

### 6.1. Peptide- and Protein-Based Vaccines

Vaccines based on peptides and proteins use fragments of human papillomavirus (HPV) proteins to stimulate an immune response. Because some of these proteins and peptides exhibit low antigenicity levels, a synergistic effect has been reported by using adjuvants [110,111].

The targeted delivery of a peptide vaccine to a particular site triggers a localized impact, suppressing tumor cells that have been activated by either specific or non-specific antigen-presenting cells (APC) [110].

These vaccines have shown immunogenic effects in CC patients, eliciting specific immune responses against HPV, and several clinical studies highlight their positive impact on precancerous lesion regression and HPV-associated cancer [112,113,114,115].

### 6.2. Live Vector Vaccines

Bacterial or viral live vectors produce numerous copies of the specific antigenic gene or protein inside the host, and the quantity of these copies is contingent on the size of the delivered molecule [110].

Vaccines based on live vectors like Listeria monocytogenes and Lactococcus lactis are employed as adjuvants to enhance HPV vaccine effectiveness. These live vectors can trigger an immune response by antigen processing in macrophages. A clinical study showcasing the efficacy of a Listeria monocytogenes-based vaccine in CC patients demonstrated positive effects in 40% of the patients [116,117].

### 6.3. Nucleic Acid-Based Vaccines

Vaccines based on nucleic acids, such as DNA and RNA, aim to induce a specific immune response against HPV. These vaccines have exhibited immunogenic effects in patients with precancerous lesions, stimulating both cellular and humoral responses [16].

The DNA vaccine pNGVL4aCRT-E7 has been utilized in multiple clinical trials aimed at treating women diagnosed with Cervical Intraepithelial Neoplasia (CIN) 2–3. The calreticulin-related plasmid DNA vaccine demonstrated an immune response in 69% of patients, with observed reduced severity in the targeted local tissues or organs after administration [118].

### 6.4. Whole Cell-Based Vaccines

Cell-based therapeutic vaccines have the potential to trigger a reversal of HPV-related diseases. These vaccines work by extracting and eliminating cells like T lymphocytes or DCs from infected tissue or pathological samples [119]. Biopsy tissues derived from the tumor and vascular system are cultured in laboratory settings, where they are altered to produce immunomodulatory cytokines. After this modification, these cells are then injected back into the host’s body, aiming to induce regression of the infection [120,121,122].

Overall, these varied HPV vaccination strategies have shown immunogenic effects and the ability to induce specific responses against the virus in patients with precancerous lesions and CC. However, further clinical studies are required to evaluate the long-term efficacy and safety of these vaccines.

Even though these vaccines have been widely used as a prophylactic and therapeutic strategy versus HPV infections, people still get HPV infection. That raises the question: Is the vaccination as effective as we thought it would be? What factors are influencing this outcome?

## 7. Impact of HPV Vaccination in LA and Challenges in Its Implementation

Over the 15 years since its introduction, the global uptake of the HPV vaccine has been lower than anticipated, which may be influenced by various factors including public perception of vaccine safety. In LA, the coverage of HPV vaccination is below expected levels, with countries such as Brazil, Mexico, and Argentina reporting significant declines in vaccination rates. This situation is further complicated by varying vaccination policies and access across the region. Additionally, the adherence to the previously recommended three-dose regimen has been challenging, though recent WHO recommendations now endorse a one or two-dose schedule for specific age groups, which could potentially improve vaccine accessibility and compliance [123].

A major concern of parents regarding HPV vaccination is the belief that it will affect their children’s sexual life, promoting its beginning at younger ages or leading to unsafe sexual practices [124]. A study was performed in Brazil evaluating the effect of the vaccination on risky sexual behavior. The results showed no significant effects concerning the onset of sexual activity or condom use during the first sexual encounter. The study’s conclusion highlighted that the HPV vaccine did not impact the initiation of sexual activity or the use of condoms. Furthermore, the need to intensify awareness campaigns aimed at parents was suggested to enhance vaccination coverage [125].

Although LA has made progress in introducing the HPV vaccine, it still faces challenges in achieving high coverage and strengthening monitoring, evaluation, and reporting systems. Improvements are necessary in monitoring systems for HPV vaccine coverage and enhancing the availability of data regarding the vaccine’s impact in the region. Challenges persist in terms of resources, knowledge, and acceptance of the HPV vaccine in LA [126].

A qualitative study performed in the Dominican Republic identified four main barriers for the screening and treatment of CC in that country; they concluded that these barriers are: limited public awareness of HPV and CC, stigma and cultural beliefs, doubts about access to and the quality of diagnostic and treatment services and losses to follow-up [17].

The barriers hindering participation in CC detection encompass a lack of public awareness regarding HPV and CC, alongside associated stigma and fear, resulting in the avoidance of testing. Knowledge barriers lead to misconceptions and hinder proactive preventive measures. Competing priorities and healthcare utilization barriers cause neglect of personal health, delays in seeking medical attention, and limited access to healthcare services. Additionally, follow-up and treatment barriers in the Dominican Republic involve distrust in healthcare quality, financial constraints, and insufficient access to medications, affecting the adherence to recommended protocols. These multifaceted barriers significantly impede timely detection, treatment, and follow-up care in CC management [18,19].

In Trinidad and Tobago, barriers and facilitators of the HPV vaccinations were also explored, agreeing to the necessity for educational campaigns promoting HPV vaccination among adolescents. Some participants expressed concerns over limited information on vaccine efficacy and safety, raising the recommendation to broaden the access to information regarding the HPV vaccine and focus on healthcare provider training to effectively advocate for vaccination. Additionally, it suggests exploring physicians’ perspectives and culturally sensitive campaigns to enhance vaccine acceptance among males. Identifying barriers and facilitators is crucial for crafting impactful vaccination campaigns [127].

Feasible solutions to combat the HPV challenge in LA involve targeted educational campaigns for healthcare providers and parents, relying on trusted sources such as doctors and schools while dispelling safety concerns through accurate information dissemination. Tactics include extending vaccination appointments for parental education, bundling the HPV vaccine with childhood immunizations and reducing financial barriers via initiatives like PAHO’s Revolving Fund. Studies emphasize the cost-effectiveness of HPV vaccination, urging its integration into public health insurance for wider access, requiring sustainable financing for successful implementation. Implementing these strategies is crucial for significantly mitigating the HPV issue in LA [19].

An attempt to overcome these barriers can also be fostering the education of the population. In this matter, a study performed in the Dominican Republic found key facilitators of HPV vaccine implementation as follows: acceptance of vaccines, established vaccination norms, accessibility of vaccination options, priority of CC concern. These facilitators collectively indicate a positive attitude towards vaccines and an existing infrastructure for vaccine delivery. This supports the potential successful implementation of an HPV vaccine program in the Dominican Republic [18].

A study performed on a Chilean population concludes that family engagement in discussions about sexual education plays a critical role in decision-making processes, particularly concerning preventing HPV infection. These conversations, however, often reflect traditional gender roles, with some parents actively participating while others do not. The lack of paternal involvement might stem from limited time, paternal disapproval, or the perception that sexual education is solely the mother’s responsibility. To effectively prevent HPV infection, it is essential to encourage open and inclusive discussions within families about sexual health. Overcoming traditional gender roles and fostering equal participation in these conversations can significantly contribute to raising awareness and ensuring comprehensive education about HPV and its preventive measures [128].

The eradication of cervical cancer relies on collaborative endeavors involving governments, civil society groups, industry, researchers, and international bodies. Ensuring the widespread availability of our efficient tools for immunization, screening, and treatment is of utmost importance. This call to action presents a chance for eliminating cervical cancer; it is our generation’s duty to guarantee that no woman succumbs to this preventable illness [20].

Unfortunately, as a consequence of the COVID-19 pandemics, a significant decrease in global online search interest in cervical cancer care was observed, reflective of the lower cervical cancer screening rates during this time [129].

Ensuring the long-term financial sustainability of HPV vaccination programs hinges on the establishment of self-financing structures with robust legal and institutional support, as well as the extension of vaccination policies providing free vaccination to children and adolescents. Although the introduction of HPV vaccines has marked significant progress in LA and the Caribbean, enhancing vaccination coverage remains imperative in order to meet both regional and global targets for cervical cancer control, aiming to reduce new cases and deaths by one-third by 2030 [12].

### 7.1. Current Status of HPV Vaccination Programs in Latin American Countries

By the close of 2023, 47 countries and territories across Latin America and the Caribbean had incorporated HPV vaccines into their national immunization schedules, extending access to adolescent girls in 89% of PAHO Member States. The adoption of HPV vaccination began in the United States in 2006, later followed by Panama and Canada in 2008. Subsequently, between 2011 and 2015, 14 additional countries in the region introduced HPV vaccines, and from 2016 onward, another 27 introductions occurred [12,130].

Currently, 42 nations in the region use the quadrivalent HPV vaccine, while one country employs the bivalent vaccine, as per the available data. The implementation of HPV vaccines in this region has markedly improved vaccine availability and accessibility since the late-2000s. The majority of the Latin American countries use the tetravalent vaccine; however, some countries such as Paraguay and Argentina could prevent around 38% of HPV with the nonavalent vaccine (Table 1), which is significantly higher than expected with the quadrivalent vaccine (13% and 22.5%, respectively) [84,87]; therefore, the nonavalent vaccine is expected to be available in the upcoming years. Unfortunately, despite these endeavors, the region is falling behind the goal of achieving 90% coverage with two doses of the HPV vaccine for girls by age 15 by 2030 [12,130].

Vaccination rates for HPV in girls aged 15 have seen only marginal increases in recent years, reaching 58% in 2019 and 61% in 2020. If this sluggish pace continues, it is projected that the region may barely attain an 80% full HPV vaccination coverage of 15-year-old girls by 2030. Notably, these estimates may not fully capture the success of programs with new HPV vaccine introductions and could underestimate achievements due to challenges in accurately recording and reporting vaccination data. The PAHO Revolving Fund has been instrumental in improving vaccine supply and financing in the region by leveraging stronger purchasing power and negotiating lower prices for the HPV vaccine [12,130].

### 7.2. Effectiveness of Vaccination Campaigns and Cervical Cancer Incidence

A clinical study assessed the safety and immunogenicity of the AS04-adjuvanted HPV-16/18 vaccine in 4- to 6-year-old girls across Colombia, Mexico, and Panama over a 36-month period post-vaccination. The results indicated a sustained immunological response with detectable antibody levels against HPV-16 and HPV-18 in all girls, high antibody concentrations at 36 months, and no withdrawals due to adverse events. Due to the vaccine’s well-tolerated nature and its ability to induce a sustained high immunological response, it has been proposed as a pediatric HPV vaccination strategy to overcome limitations in adolescent vaccination programs. HPV vaccine may be given at the same time as other vaccines. Concomitant administration of HPV vaccine with other vaccines in the children’s vaccination program would minimize the number of visits required to deliver each vaccine individually [131].

Latin American countries are also concerned about the economic impact HPV-related diseases represent. The findings of a report underscore a critical economic imperative to prevent HPV-related diseases in Peru. The average treatment cost per patient of 59.9 USD and the total annual cost of treating genital warts amounting to 25.1 million USD reflect a substantial financial burden. Surgical excision emerged as the most expensive treatment method, while trichloroacetic acid (TCA) stood as the most cost-effective option. Notably, human resources constituted the largest proportion of costs across all treatment alternatives. This study reveals the economic impact of genital warts in Peru, yet the conservative cost estimates, excluding all healthcare providers and out-of-pocket expenses, emphasize the substantial financial burden. It underscores the urgent need for preventive measures due to the preventable nature of this condition through vaccination [132].

In LA women with various cervical lesions, HPV16 has been proven to be the predominant genotype across all grades, particularly prominent in cervical cancer cases. HPV16 frequency increased significantly with lesion severity, while other high-risk HPV types did not exhibit similar trends in cancerous lesions versus less severe ones. This highlights the importance of HPV16 across the clinical spectrum and its association with neoplastic lesions, particularly in CIN3 and CC. The latter provides insights into the effectiveness of the nonavalent vaccine against high-risk HPV genotypes in cases of CC. Most women with CC had single infections of high-risk HPV, emphasizing the significance of HPV16 in the development of cervical lesions. The forementioned demonstrates the need for the implementation of efficient and sustainable screening programs to complement HPV vaccination efforts, which supports the use of HPV DNA detection as a screening method over traditional cytology, considering regional variations in genotype distribution [88].

### 7.3. Current Actions by LA Countries

In LA, cervical cancer (CC) rates remain high despite the introduction of cytological screening programs. Studies conducted in the region have made significant contributions to understanding HPV’s impact. They revealed that high-risk HPV types like HPV-16 and HPV-18 are prevalent, causing most CC cases and being classified as group 1 carcinogens, emphasizing the need for prevention strategies, showcasing the integration of HPV vaccines in national programs as crucial for reducing pre-cancerous cervical lesions. Furthermore, research in LA highlighted the effectiveness of HPV testing in screening for cervical cancer, offering valuable insights for better screening programs [133].

In Colombia, ensuring equitable access to immunization is a national priority, marked by strategic initiatives within public health planning. Programs like the 2018–2022 National Development Plan and the Ministry of Health’s Observatory for Health Inequality Measurement and Equity Analysis aim to monitor and address health disparities arising from social disadvantages. The Expanded Program on Immunization (EPI) uses multifaceted approaches to identify gaps in coverage, employing rapid monitoring and collaborating with PAHO for capacity-building activities. Recent efforts involved workshops to enhance analytical capabilities in assessing social inequalities in immunization, focusing on subnational data and engaging with underprivileged communities to improve awareness and accessibility [134].

Colombia has actively developed tools like the National Immunization Equity Booklet, jointly designed by the Ministry of Health and PAHO, to inform equitable immunization program planning. This resource identifies disparities in routine vaccine coverage across municipalities and aids MSPS personnel in understanding and improving immunization status. Despite challenges in implementing equitable monitoring, Colombia’s commitment to partnerships, technical capacity enhancement, and the development of equity-centered resources showcases its leadership in advancing immunization equity efforts [134].

In Bolivia, community engagement and cross-sectoral cooperation have been pivotal in bolstering HPV vaccine accessibility and acceptance. The Family, Community, and Intercultural Health Model (SAFCI) has notably enabled public participation in supervising and managing the healthcare system, particularly in promoting HPV immunization [134].

Sucre, Peru, exemplifies effective collaboration between local health and education authorities, streamlining school-centered vaccination initiatives. Information sharing among municipal bodies, educational institutions, health centers, and local health committees has streamlined campaign planning and ensured precise student enrollment records. This joint effort involves parental guidance sessions, coordinated vaccination schedules, and impactful social mobilization strategies alongside the School Board Association to encourage vaccine uptake among peers. Despite challenges in attaining accurate vaccination metrics due to census data concerns, Sucre’s collaborative approach and community partnerships have notably mitigated these obstacles [134].

Bolivia’s success underscores the importance of evidence-driven decision-making, multi-sector partnerships, and community involvement in reinforcing immunization equity. The SAFCI model stands out for fostering community engagement and has been instrumental in the successful HPV vaccine implementation countrywide [134].

## 8. Future Directions and Recommendations

CC is a threatening disease that, despite all the measures different countries have taken to eradicate it, continues to be among the foremost in terms of prevalence and mortality in Latin American countries. Undoubtedly, there is still a long way to go, but what criteria should be considered when making decisions?

CC is a disease the main causal factor of which is HPV infection; however, it is well known that the infection itself is incapable of inducing it, which is why the presence of other cofactors is necessary. Within these cofactors, we can find genetics and environment. In both cases, it is necessary to continue with research on how they can influence the evolution of the disease. On the other hand, understanding the molecular aspects can help us develop better treatment and prevention strategies, allowing us to reduce the evolution of the infection towards premalignant lesions.

In terms of environmental factors, it is important to continue with the sexual education of the population, promoting the use of condoms, and promoting healthy habits (such as avoiding smoking), healthy eating, and sports, which can impact not only the development of CC but also the development of other diseases.

It is important to continue with campaigns that eliminate the taboo for women regarding check-ups, especially in regions with limited access to information; a study conducted in Italian women reports that only 15% of participants were able to draw vulvar anatomy correctly, and 61% approached their genitals with feelings of embarrassment [135]. In LA, these data have not been explored, but there is knowledge that sexual education must be regulated more efficiently in each country with an adequate teaching–learning process [136]. The teaching should include tools that encourage self-exploration; for example, vulvar self-examination could contribute to the prevention of HPV-related diseases [135].

In short, screening campaigns for the detection of cervical lesions and HPV are essential tools that permit early diagnosis. In addition, they help to detect the prevalence and distribution of the different HPV genotypes, the presence of which allows for predicting the development or evolution of the disease; it will undoubtedly continue to be the primary defense against CC. Additionally, the utilization of self-collection for cervical screening can improve screening strategies by increasing accessibility, convenience, and women’s participation, ultimately leading to better overall detection and prevention of the disease [137].

Vaccination campaigns present a promising solution for CC; so far, it has been found that 47 member countries of the PAHO have introduced some of the HPV vaccines in their vaccination campaigns. However, it has been seen that the coverage of vaccines does not exceed 60% of the population in more than half of them, which is why it can be considered a strategy that is still under construction [130].

The vaccination schedules implemented in most Latin American countries involve two doses. In addition, the most used vaccine is the quadrivalent, followed by the bivalent [12]. However, in studies carried out in a young, sexually active population prior to vaccination, it was reported that the coverage of the nonavalent vaccine would be better for preventing genotypes distributed in the population [84,85,86,87]. So, in future strategies, it might be worth including it in vaccination schedules.

The male population plays a fundamental role in the development of CC since it has been seen that men present a higher percentage of positive samples compared to women. In addition, it has been seen that they present many genotypes that cover current vaccines [97,98], so including them in the vaccination plan could be a favorable situation in the fight against CC.

A characteristic that can significantly impact the disease’s control is that the prevalence and frequency of HPV genotypes show very characteristic patterns in each country and region, so the most frequent HPVs in one country may not be the same in another or within the same country’s subregions (Table 1). In the same way, they can vary with age; the prevalence and variability of HPV genotypes are usually greater in a young population, while the variability in the genotypes present decreases as age increases [88]. Additionally, it has been seen that the patterns of genotypes present in a population could change over the years [98].

The characteristic pattern of prevalence and genotype distribution in each region should be taken into account to carry out appropriate vaccination strategies, since although HPV16 appears to be one of the most frequent in all study groups and is preventable with the bivalent and tetravalent vaccines, which are primarily implemented in Latin American countries, other HR-HPVs not included in vaccines could induce the development of CC. Additionally, it is expected that, after vaccination, the distribution of genotypes will be modified in the different study regions, so monitoring the prevalence and genotype HPV distribution should be taken very seriously; in this way, this action can deal with surprises that may arise after vaccination.

Perhaps one of the most important challenges is the fact that there is still a lot of misinformation, stigma, and cultural beliefs regarding HPV vaccination, so work definitely needs to continue on the dissemination of information related to infection by HPV and CC and the benefits that the application of the vaccine can bring; this information must be understandable by both parents and adolescents so that over time the barrier of beliefs and collective fear decreases.

## 9. Conclusions

In conclusion, despite significant strides in understanding and combatting HPV and cervical cancer, the global impact of this disease persists, underscoring the urgent need for ongoing efforts in prevention, treatment, and vaccination. Hormonal factors, particularly estrogen and progesterone, play a substantial role in cervical cancer development, presenting potential therapeutic targets. The global implementation of HPV vaccines has been a crucial advancement, yet challenges persist—especially in LA—affecting vaccination rates. Addressing these challenges through educational campaigns, healthcare provider training, and community engagement is essential. While progress is evident, sustained efforts, collaborative initiatives, and comprehensive strategies are imperative for achieving widespread vaccination coverage and ultimately eliminating HPV-related diseases.

## Figures and Tables

**Table 1 viruses-16-00327-t001:** HPV distribution reported in some Latin American countries.

Country and Population	Study Characteristics	HPV Prevalence	HPV Genotypes More Frequents in Positive Samples		Single Infection (SI) or Multiple Infection (MI) Data	Observations	Ref.
**Section A: Studies carried out in women with a diagnosis of some degree of cervical lesion or CC**							
20 states of Mexico Mexico	Women from 18–90 years screened for CC. HPV genotyping Sampling: Liquid-based cytology Disease stage: Cytology. HPV detection by qPCR BD Onclarity HPV Assay N = 60,135	24.78%	Prevalence in all states: HPV 16: 4.13% HPV 31: 4.12% HPV 51: 3.39% HPV52: 3.29% HPV18: 1.70%	NILM HPV16: 3.2% HPV31: 3.4% HPV51: 2.6% HPV52: 7.77%	N.D.	HPV prevalence and genotypes frequency varied across the different states analyzed. ↑ prevalence (47.79%) in W under 25 years ↑ HR-HPV prevalence in W without social security	[93]
				HSIL HPV16: 48.8% HPV31: 13.3% HPV51: 5.5% HPV52: 2.75%			
Aguascalientes, Colima, Guanajuato, Jalisco, Michoacan and Nayarit. (Western of Mexico) Mexico	Women aged 18–82 years Groups: Open Population (OP) N = 3000 CIN1 N = 77 CC. N = 96 HPV genotyping Sampling: Liquid-based cytology Disease stage: Histology. HPV detection by LA or COBAS 4800 test.	OP: 12.1% CIN1: 53.3% CC: 77.1%	CIN 1: HPV16: 23.7% HPV66: 23.7% HPV6:21.1%, HPV53: 18.4% HPV59: 15.8% HPV89: 15.8% HPV51:13.2% HPV56:13.2% HPV18:10.5% HPV39: 10.5%	CC: HPV16: 50% HPV18: 18.9% HPV59: 14.9% HPV11: 10.8% HPV45: 9.5% HPV58: 9.5%	SI in OP positive samples: 32.7% MI in OP: 67.3% MI in CIN1: 77.5% MI in CC: 75.7%	HPV genotypes varied across the different geographical regions that belong to the western Mexico.	[95]
				OP: HPV16: 22.0% HPV59: 18.0% HPV66: 16.3% HPV52: 15.3% HPV51: 15.0% HPV31: 14.3%			
Sao Paulo Brazil	Women aged 14–95 years. HPV genotyping Sampling: Liquid-based cytology Disease stage: Cytology HPV detection by HC2 or Cobas and PapilloCheck test N = 665	48.6%	HR-HPV HPV16: 23.2% HPV56: 21.0% HPV52: 8.7% HPV31: 7.7% HPV53: 7.7% HPV51: 7.4%	LR-HPV HPV42: 12.1% HPV6: 6.2% HPV44: 4.3% HPV43: 4% HPV40: 2.8% HPV11: 1.5%	SI: 65% MI. 2HPV: 18% 3HPV: 11% 4HPV: 4% ≥5HPV: 2%	No association between LR-HPV and HR-HPV types with age group was observed.	[92]
			The most frequent types in: **NILM:** HPV56: 19.1% HPV16: 17.7% **HSIL:** HPV16: 37.2% **ICC**: HPV16: 66.7%				
Argentina, Colombia, Costa Rica, Honduras, Paraguay, and Uruguay ESTAMPA Study.	Women aged 30–64 years HPV genotyping Sampling: Liquid-based cytology Disease stage: Histology. HPV detection by BSGP and RLB N HPV detection = 27,558 N HPV genotyping = 1252	13%	≤CIN1 HPV16: 14.5% HPV52: 11.1% HPV31: 10.3% HPV56: 9.5% HPV59: 8.2% HPV58: 7.3% HPV66: 7%	CIN2 HPV16: 19.8% HPV52: 15.7% HPV18: 11.6% HPV31: 11.6% HPV58: 9.1% HPV35: 8.3% HPV51: 8.3%	SI in:≤CIN1: 57.1% CIN2: 57% CIN3: 72.2% CC: 91.6% MI by 2 HPV: ≤CIN1: 16.2% CIN2: 19.8% CIN3: 18.6% CC: 4.8% MI by ≥3 HPV: ≤CIN1: 4.7% CIN2: 5% CIN3: 4.6% CC: 1.2%	The risk of HPV16 infection in CIN3 cases increased with age. HPV16 was the most prevalent genotype in all histological groups and show a significant increase with histological grade	[88]
			CIN3 HPV16: 51.5% HPV31: 12.9% HPV52:11.3% HPV58: 8.8% HPV33: 8.2% HPV18: 7.2%	CC HPV16: 65.1% HPV45: 8.4% HPV18: 7.2% HPV52: 4.2% HPV58: 4.2%			
Rio de Janeiro city (RJ) and Belem city Para State (PA) Brazil	Patients diagnosed with CC HPV genotyping Sampling: Biopsy Disease stage: Histology. HPV detection by DNA sequencing and High + Low Papillo-mastrip N = 1183	N.D.	RJ: N = 590 HPV16: 61.8% HPV18: 14% HPV45: 6% HPV35: 2.2% HPV31: 2.1% HPV33: 2.1% HPV58: 1.9%	PA: N = 593 HPV16: 62.1% HPV18: 10.9% HPV33: 4.1% HPV45: 4.1% HPV31: 3% HPV52: 2.7% HPV35: 2.5%	SI in RJ samples: 96.6% SI in PA samples: 95.4%	HPV 16 is present in most cases of MI HPV16 (+) in 40/41 tumors HPV18 (+) in 25/41 tumors	[94]
**Section B: Studies carried out prior to vaccination**							
Asuncion Paraguay	Unvaccinated women aged 18–25 years HPV genotyping Sampling: Liquid-based cytology HPV detection by CLART HPV2 test N = 208	54.8%	HPV58: 14.9% HPV16: 12.3% HPV51: 12.3% HPV66: 12.3% HPV52: 11.4% HPV53: 11.4%		SI: 57% MI. 2HPV: 30.1% 3HPV: 7% ≥4HPV: 5.3%	42.3% of W HR-HPV (+). Prevalence of HPV preventable with vaccines: -Bivalent: 8.2% -Quadrivalent: 13.0% -Nonavalent: 38.0%	[84]
Berazategui, Posadas, La Banda, and Buenos Aires Argentina.	Unvaccinated sexually active women aged 15–17 years HPV genotyping Sampling: Liquid-based cytology HPV detection by BSGP and RLB N = 957	56.3%	HPV16: 11.1% HPV52: 10.8% HPV56: 8.3% HPV51: 7.4% HPV58: 7.3% HPV31: 7.1%		SI *: 20.0% MI *: 2HPV: 14.3% 3HPV:10.1% 4HPV: 5.1% ≥5HPV: 6.8%	42.2% of W HR-HPV (+). Prevalence of HPV preventable with vaccines: -Bivalent: 15.2% -Quadrivalent: 22.5% -Nonavalent: 38.5%	[87]
Soacha, Girardot, and Manizales. Colombia	Unvaccinated sexually active women aged 18–25 years HPV genotyping Sampling: Liquid-based cytology HPV detection by LA N = 1782	60.3%	HR-HPV HPV16: 11.28% HPV52: 7.91% HPV51: 7.86% HPV58: 7.07% HPV59: 7.01% HPV39: 5.72% HPV31: 4.55%	LR-HPV HPV53: 7.0% HPV89: 6.3% HPV61: 6.1% HPV66: 6.0% HPV62: 5.9% HPV84: 5.9% HPV73: 4.8%	SI in positive samples: 38.5% MI in positive samples: 61.5%	42.2% of W HR-HPV (+) 44.4% of W LR-HPV (+) Prevalence of HPV preventable with vaccines, was: -Bivalent: 14.4% -Nonavalent: 30.75%	[86]
26 state capitals and the Federal District of Brazil. Brazil	Unvaccinated women (W) and men (M) aged 16–25 years HPV genotyping Sampling: Liquid-based cytology HPV detection by LA. N = 6388	Total (W + M): 53.6% Women: 54.6% Men: 51.8%	Total (W + M): HPV52: 7.8% HPV16: 7.5% HPV62: 6.8% HPV89: 6.3% HPV61: 6.0%	Women *: HPV16: 8.9% HPV 52: 8.8%	SI: 42.2% MI. 2 HPV: 26.7% 3HPV: 12.2% >4HPV: 18.8%	38.6% of W HR-HPV (+) 29.2% of M HR-HPV (+) Prevalence of HPV preventable with vaccines, was: -Quadrivalent: 14.8% -Nonavalent: 27.7%	[85]
				Men *: HPV59: 6.5% HPV52: 6.0%			
Guatemala City and Puerto Barrios Guatemala	Asymptomatic sexually active women HPV genotyping Sampling: Liquid-based cytology HPV detection by H13 test N = 1717	13%	HPV16: 22% HPV39: 11% HPV18: 11% HPV58: 10% HPV52: 8%	HPV45: 8% HPV59: 7% HPV68: 5% HPV35: 4% HPV56: 4%	MI: 22%	↑ prevalence (21%) in women <30-years age	[90]
**Section C: Other miscellaneous studies**							
Havana (Ha) Villa Clara (VC), and Holguín (Ho) Cuba	Women aged 16–67 HPV genotyping Sampling: Liquid-based cytology HPV detection by CLART HPV 2 kit N = 500	Total: 4.8% Ha: 18% Ho: 15% VC: 13%	HPV16: 23% HPV31: 10.8% HPV33: 8.1% HPV53: 8.1% HPV61: 8.1% HPV66: 8.1%		MI in positive samples: 23%	79.7% of W HR-HPV (+). 27% of W LR-HPV (+) ↑ MI in W under 25 years. HPV genotypes varied across the different regions	[89]
Bogotá Colombia	Women with cervical intraepithelial lesions and their sexual stable partners. HPV genotyping Sampling: Liquid-based cytology HPV detection by LA N = 25 partners (25W and 25M)	Women: 80% Men: 56%	Women HPV16: 25% HPV45: 15% HPV54: 15% HPV62: 15%	Men HPV83: 28.5% HPV16: 21.4% HPV62: 21.4% HPV68: 21.4%	MI in Women: 30% MI in Men: 64%	Men with MI show a history of more than five sexual partners in their lifetime. 28.0% of partners shown concordance in at least one HPV 60% of the studied couples showed discordant results	[96]
Guayaquil Ecuador	Men and women 18–70 years HPV genotyping Sampling: Liquid-based cytology HPV detection by GA N = 800 (400 M and 400 W)	Total (W + M): 51.38% W: 39.5% M: 63.5%	Women: HPV39: 17.09% HPV16: 13.92% HPV6: 13.29% HPV58: 10.76%	Men HPV6: 35.18% HPV16: 7.39% HPV18: 10.67% HPV11: 10.28%	SI in women: 54.4% SI in men: 62%	Positive men samples show infection by HPV genotypes covered by vaccines	[97]
Northwestern region of Mexico. Mexico	Asymptomatic males with a female sexual Partner diagnosed with LGSIL. HPV genotyping Sampling: Liquid-based cytology HPV detection by LCD-Array N = 1769	41.9%	HR-HPV HPV66: 13.2% HPV16: 11.1% HPV59: 9.2% HPV51: 7.9% HPV39: 6.9% HPV56: 6.5%	LR-HPV HPV6: 21.1 HPV91: 11.5% HPV42: 9.5% HPV62: 9.1% HPV90: 8.5% HPV11: 8.3%	SI * 21.4% MI *: 2 HPV: 10.4% 3 HPV: 5% 4HPV: 2% ≥5HPV: ≈2.6%	Prevalence of HPV preventable with vaccines: -Bivalent: 12.77% -Quadrivalent: 38.58% -Nonavalent: 48.39%	[98]

* The percentage was calculated by taking the total number of participants as 100% and not the rate of patients positive for HPV. The N represents the number of samples analyzed; it does not include those that for some reason were excluded from the study. Abbreviatures: Ref (reference); HPV (Human papillomavirus); LR-HPV (Low risk HPV); HR-HPV (High risk HPV); OP(open population); ICC (invasive cervical cancer); NILM (Negative for Intraepithelial Lesion or Malignancy); LGSIL (low-grade squamous intraepithelial Lesion); HSIL (High-grade squamous intraepithelial Lesion); CC (Cervical Cancer); CIN (Cervical Intraepithelial Neoplasia); RJ (Rio de Janeiro city); PA (Belem city Para State in Amazonian region); BSGP (Broad-Spectrum General Primers); RLB (Reverse Line Hibridization); LA (Linear Array HPV Geno-typing Tests); HC2 (Hybrid Capture 2); GA (GenoArray Diagnostic Kit) M (Men); W (Women); SI (single infection); MI (Multiple infection); ↑ (Increase ); (+) (positive); N.D. (Not available Data).

## Data Availability

Not applicable.

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
