# Peer review of "HPV and Cervical Cancer: Molecular and Immunological Aspects, Epidemiology and Effect of Vaccination in Latin American Women"

_viruses, 2024, doi:10.3390/v16030327_

Round 1

Reviewer 1 Report

Comments and Suggestions for Authors This could be an interesting review article.

In my opinion, the Authors should reword it, i.e. first describe the characteristics of HP viruses and prevention - types of vaccines. The second part should present the epidemiological situation in Latin America and then the methods and possible effects of specific prevention introduced in this region.

The article seems to benefit from some clarity.

Author Response

Dear Reviewer, we are truly thankful for your comments and suggestions. After having taken them into consideration, proper modifications were made and hereby we respond: 

Reviewer Comment 

This could be an interesting review article. In my opinion, the Authors should reword it, i.e. first describe the characteristics of HP viruses and prevention - types of vaccines. The second part should present the epidemiological situation in Latin America and then the methods and possible effects of specific prevention introduced in this region. The article seems to benefit from some clarity 

Response:  

  • We have thoroughly reviewed your suggestion, and we find merit in reorganizing our content to enhance clarity and impact. Our proposed revised structure involves initially providing a molecular background on HPV, followed by an exploration of the hormonal and immune environments. This sequential approach aims to bolster the effectiveness of our vaccination explanation and subsequently facilitate a more comprehensive understanding of the Latin American situation. We believe this revised order of ideas will significantly enhance the overall coherence and persuasiveness of our presentation. 

We would be grateful to receive your feedback and hopefully your positive approval of this manuscript 

Reviewer 2 Report

Comments and Suggestions for Authors

Dear authors,

this is an interesting paper on HPV and cervical cancer in Latin America. Study design is appropriate but some revisions are needed to improve the manuscript.

Some more lines should underline the risk of other HPV related neoplasias, as in 10.1186/s12885-020-07452-6.

Also further improvement in clinician training and in women self -awareness are needed to reduce the burden of the disease an other HPV related neoplasia (as in 10.1097/LGT.0000000000000777 and in 10.1097/LGT.0000000000000585)

Finally, some words should be spent on vaginal and vulvar HPV related intraepithelial neoplasia ( as in 10.1136/ijgc-2022-004213 and in 10.1136/ijgc-2021-003262)

I thank the authors for their precious work

Comments on the Quality of English Language

Minor

Author Response

Dear Reviewer, we are truly thankful for your comments and suggestions. After having taken them into consideration, proper modifications were made and hereby we respond:

Reviewer Comment

Dear authors, this is an interesting paper on HPV and cervical cancer in Latin America. Study design is appropriate, but some revisions are needed to improve the manuscript. Some more lines should underline the risk of other HPV related neoplasias, as in 10.1186/s12885-020-07452-6. Also further improvement in clinician training and in women self -awareness are needed to reduce the burden of the disease an other HPV related neoplasia (as in 10.1097/LGT.0000000000000777 and in 10.1097/LGT.0000000000000585) Finally, some words should be spent on vaginal and vulvar HPV related intraepithelial neoplasia ( as in 10.1136/ijgc2022-004213 and in 10.1136/ijgc-2021-003262) I thank the authors for their precious work

Response:

  • We express our sincere gratitude for your valuable comments and insightful suggestions. Following a thorough review, we have diligently incorporated the pertinent information from some articles that you recommended seamlessly into our text. We firmly believe that these additions have significantly enriched and enhanced the comprehensiveness of our ideas. Your guidance has been instrumental in refining our work, and we extend our heartfelt thanks for your constructive input. You can view the information inserted in the document highlighted in green.

We would be grateful to receive your feedback and hopefully your positive approval of this manuscript

Reviewer 3 Report

Comments and Suggestions for Authors

Dear authors,

According to my opinion, the text is too extensive and sometimes complex. I recommend shortening some parts. In particular, table 1, which is confusing, must be significantly modified. I recommend to reconsider the article after major revision.

Another comments

 (see attached file)

Lines 33-35

Human Papillomavirus (HPV) poses a significant global health challenge due to its strong link with cervical cancer (CC), a leading cause of women's mortality worldwide [1].

+ Lines 59-60

In fact, CC ranks as the second most common cancer in women globally[1].

Comment: Are you sure? According to your quote and according to Sung et al (see below), CC is the fourth most common cancer in women worldwide (both in incidence and death). Moreover, the article is about the situation in Latin America, so the incidence and mortality in this region should be given. In the Part Nr 5. “HPV and Cervical Cancer in Latin America“ you report number of cases and not the incidence of the disease.

Sung H et al. Global Cancer Statistics 2020: GLOBOCAN Estimates of Incidence and Mortality Worldwide for 36 Cancers in 185 Countries. CA Cancer J Clin. 2021 May;71(3):209-249. doi: 10.3322/caac.21660. Epub 2021 Feb 4. PMID: 33538338.

Lines 41-44

The HPV lifecycle begins with the virus interacting with host cells, particularly keratinocytes in the body's layered tissues. HPV capitalizes on minor tissue injuries, such as those during sexual activity, binding to host cells and starting its replication and transcription process [3].

Comment: The truth is that HPV infection is transmitted most often during coital and non-coital sexual activities. However, skin-to-skin or skin-to-mucosa contact is sufficient to transmit HPV infection, e.g. when washing children or also by autoinoculation in the same person. Therefore, this statement should be slightly reformulated.

Lines 83-87

According to the International Agency for Research on Cancer (IARC), out of the more than 200 known HPV genotypes, some are classified as high-risk HPV (HR-HPV), being the most common HPV-16 and HPV-18. These genotypes represent approximately 70% of the cases, but the list includes at least ten others, such as 31, 33, 35, 39, 45, 51, 52, 86 56, 58, and 59 [3,4,17].

Comment: Probably is better to cite the original article.

Bouvard V, Baan R, Straif K et al. A review of human carcinogens--Part B: biological agents. Lancet Oncol 2009; 10(4): 321-322.

Lines 215-216

In 2002, an IARC case-control study showed a strong relationship between oral contraceptive use and duration of use [46].

Comment: It does not make sense. Rewrite e.g. as follows:  In 2002, an IARC case-control study showed a strong relationship between duration of oral contraceptive use and a risk of cervical cancer in women with cervical HPV infection [46].

You should also state whether it was a cyclical oral contraceptive or a progestin-only contraceptive or both.

Lines 288-289

Studies carried out in young, unvaccinated, and sexually active Latin American women reported a high prevalence of HPV (greater than 50%);

Comment: What does it mean young? Under 25 years?

Lines 320-326

Regarding the distribution of the other HPVs, their behavior is very unpredictable in the different countries and even in the different regions into which each country is divided; for example, a study carried out in Cuba found differences in the number of coinfections in each of the three regions evaluated, as well as changes in the frequency of the different HPV genotypes in which Holguin stands out for having a lower percentage of coinfections compared to Havana and Villa Clara, as well as having a marked presence of HPV 89, which is not important in Havana [81].

Comment: The sentence is too long and confusing. Please rephrase and clarify.

Table 1

Comment: Table 1 is very confusing, it must be restructured. No one will study such a difficult table.

Example for changing the table.

Author, year of publication

Region

Number of subjects

Type of detection

Overall HPV

prevalence

NILM

CIN1/LSIL

CIN 2

HSIL

(cytology or histology?

CC

Campos-Romero, 2019

Mexico (20 states)

N = 60,135

PCR

HPV 16: 4.13%

HPV 31: 4.12%

HPV 51: 3.39% HPV52: 3.29% HPV18: 1.70%

HPV16: 3.2% HPV31: 3.4% HPV51: 2.6% HPV52: 7.77%

NA

NA

HPV16: 48.8% HPV31: 13.3% HPV51: 5.5% HPV52: 2.75%

NA

etc

NA – not available

If the table is too big, you can make two tables; e.g. Table 1 will contain HPV distribution in cytology results and Table 2 may contain HPV distribution in histological diagnoses.

Lines 363-371

However, a study carried out in Colombia in heterosexual couples in which the woman presented cervical intraepithelial lesions reported that only 28% of the couples studied showed concordance in at least 1 HPV, and 60% of the couples showed discordance, that is, there was no HPV in common, which is most likely due to differences between men and women in the time to resolve the infection additionally, the percentage of co-infections was higher in man than women (64% and 30% respectively); this man with multiple infections reported having had more than five sexual partners throughout their lives, which is a factor that is related to the prevalence of HPV[89].

Comment: The sentence is too long and confusing. Please rephrase and clarify.

Lines 372-374

Martins et al. reported a random subsampling of HPV-negative samples, of which approximately 13.9% proved to be positive for some HPV genotypes.

Comment:  I do not understand this. Was it provided by other method?

Lines 383-384

Unfortunately, the incidence of diseases caused by HPV infections is still alarming, leading CC to be the second most common cancer in women globally.

Comment:  See comment to lines 59-60.

Lines 403-406

GardasilÒ (Merck), is a quadrivalent vaccine used to prevent infections by HPV-6, 11, 16, and 18, which altogether represent the vast majority (main cause) of cutaneous lesions and cervical malignancies [11]. Additionally, this vaccine has proved to be useful in reducing oral cavity, penis, vulva, and anus HPV infections [97–99].

Comment:  Bivalent HPV vaccine reduces oral HPV infection, too.

Herrero R, Quint W, Hildesheim A, Gonzalez P, Struijk L, Katki HA, Porras C, Schiffman M, Rodriguez AC, Solomon D, Jimenez S, Schiller JT, Lowy DR, van Doorn LJ, Wacholder S, Kreimer AR; CVT Vaccine Group. Reduced prevalence of oral human papillomavirus (HPV) 4 years after bivalent HPV vaccination in a randomized clinical trial in Costa Rica. PLoS One. 2013 Jul 17;8(7):e68329. doi: 10.1371/journal.pone.0068329. PMID: 23873171; PMCID: PMC3714284.

Lines 407-409

Besides the quadrivalent, there is another GardasilÒ vaccine (Gardasil 9Ò by Merck) directed to 9 HPV serotypes, including HPV-58, 53, 45, 33, 31, 18, 16, 11, and 6, covering approximately 90 % of cervical malignancies [100].

Comment:  Why is the number series in this order and not the other way around?

Lines 471-473

A major concern of parents regarding HPV vaccination is the belief that it will affect their children´s sexual life promoting its beginning at younger ages or leading to unsafe sexual practices.

Comment: Citation should be noted. Was it a questionnaire survey of parents?

Lines 485-488

A qualitative study performed in Dominican Republic identified four main barriers for the screening and treatment of CC in that country; they concluded these barriers are: Limited public awareness of HPV and CC, stigma and cultural beliefs, doubts about access and quality of diagnostic and treatment services and losses to follow-up [14].

Comment: Citation Nr. 14 (Liebermann, 2020) refers about Parent-Level Barriers and Facilitators to HPV Vaccine Implementation ant not about barriers for the screening and treatment of CC.

Lines 543-550

Notably, the theoretical coverage of the bivalent HPV vaccine reached 21.66% of high-risk HPV positive cases. These findings stress the imperative for further research and specific prevention strategies for HPV in men, particularly in the northwestern Mexican region. The study concludes that the bivalent vaccine shows a coverage of 21.66% in high-risk HPV positive cases among men in northwestern Mexico. This suggests that the vaccine could be effective in preventing HPV infections in a significant portion of the male population [88].

Comment: 1) You repeat the same.  2) This must be discussed or deleted. Authors of citation Nr 88 include genotype 66 amohg HR HPV genotypes. According to your citations Nr 3,4,17, however, there are 12 HR HPV genotypes in total. According to IARC, there are 13 HR HPV genotypes in total, but HPV 66 is not included. Moreover, your article is named “HPV and Cervical Cancer: molecular and immunological aspects, epidemiology and effect of vaccination in Latin American women“. I recommend removing all mentions of HPV in men.

Lines 575-576

Currently, 42 nations in the region use the quadrivalent HPV vaccine, while one country employs the bivalent vaccine, as per the available data.

Comment: I assume the citation is at the end of the paragraph. If no country uses the nonavalent vaccine (even for own payment by the patient), it should be clearly mentioned.

Lines 732-780

Conclusions

Comment: The Conclusion section is too long, it should be shortened. Part Conclusions should contain a summary on several lines.

Lines 1039-1072

Comment: Citation 81 is incomplete. Correct it.

Guilarte-García E, Soto-Brito Y, Kourí-Cardellá V, Limia-León CM, Sánchez-Alvarez ML, Rodríguez-Díaz AE, López-Fuentes LX, Méndez-González M, Aróstica-Valdés N, Bello-Pérez M, Pérez-Santos L, Pintos-Saavedra Y, Baños-Morales Y. Circulation of Human Papillomavirus and Chlamydia trachomatisin Cuban Women. MEDICC Rev. 2020 Jan;22(1):17-27. doi: 10.37757/MR2020.V22.N1.5. Erratum in: MEDICC Rev. 2020 Jan;22(1):35. PMID: 32327618.

Author Response

 Dear Reviewer, we are truly thankful for your comments and suggestions. After having taken them into consideration, proper modifications were made and in the attached file we respond every point.

We would be grateful to receive your feedback and hopefully your positive approval of this manuscript.

Reviewer 4 Report

Comments and Suggestions for Authors

Hernández-Silva et al. have authored a comprehensive review article that delves into the molecular and immunological aspects of Human Papillomavirus (HPV), particularly its crucial role in causing cervical cancer. The review primarily focuses on the epidemiology of HPV and cervical cancer in Latin American women, highlighting the region's increased prevalence and the diversity of HPV genotypes. Moreover, it critically evaluates the impact of HPV vaccination programs on cervical cancer rates in Latin America, noting significant success in various countries due to these vaccines' influence on immune responses. Conclusively, the article synthesizes current knowledge about HPV and cervical cancer, with a specific emphasis on Latin American women. It addresses molecular, immunological, and epidemiological aspects and assesses the effect of vaccination initiatives, contributing to the global conversation on cervical cancer prevention and emphasizing the need for ongoing efforts in this area.

The claims are properly placed in the context of the previous literature. The experimental data support the claims. The manuscript is written clearly enough that most of it is understandable to non-specialists. The authors have provided adequate proof for their claims, without overselling them. The authors have treated the previous literature fairly. The paper offers enough details of methodology so that the experiments could be reproduced.

Minor revisions

Line 2-3, Title, "HPV in Latin American Women: Cancer Links, Molecular Insights, and Vaccination Impacts"

For the "MDPI Viruses" journal, the abstract should be around 200 words maximum.

Line 16-29, Abstract, "This review addresses the relationship between Human Papillomavirus (HPV) and cervical cancer, focusing on Latin American women. It explores molecular and immunological aspects of HPV, its role in cervical cancer development, and the epidemiology in this region, highlighting the prevalence and diversity of HPV genotypes. The impact of vaccination initiatives on cervical cancer rates in Latin America is critically evaluated, noting both successes and challenges in implementation. The review synthesizes current knowledge, emphasizes the importance of continued research and strategies for cervical cancer prevention, and underscores the need for ongoing efforts in this field."

Line 30,
"Keywords: HPV Epidemiology; Cervical Cancer Prevention; HPV Vaccines; Latin America; HPV Genotypes; Immunological Aspects of HPV; Cervical Cancer Treatment Strategies; Public Health; Vaccine Implementation; HPV Infection."

Line 33-34, "Human Papillomavirus (HPV) poses a significant global health challenge due to its strong link with cervical cancer (CC), a leading cause of women's mortality worldwide" => "Human Papillomavirus (HPV) represents a significant global health challenge, particularly in its association with cervical cancer (CC), the fourth most common cancer in women worldwide. Additionally, cervical cancer ranks as the fourth leading cause of cancer-related mortality in women globally. In 2020, approximately 604,000 new cases and 342,000 deaths were attributed to this disease."

https://www.who.int/news-room/fact-sheets/detail/cervical-cancer

Line 142-145, "Approximately 10% of the patients with HPV infection will develop a persistent infection that will lead to the appearance of low- or high-grade intraepithelial lesions and subsequently lead to the development of cancer, which implies that the immune system can eliminate HPV in approximately 90% of infection cases." => "Approximately 10% of women with an HPV infection develop a persistent infection, which can potentially lead to the formation of low- or high-grade intraepithelial lesions (CIN2+). However, it is important to note that only a subset of these women with persistent infections and high-grade lesions will eventually develop cervical cancer. While the lifetime risk of acquiring an HPV infection is estimated to be around 70-80%, the actual incidence of cervical cancer without screening is much lower, affecting only about 2-3% of all women. This indicates that the immune system is capable of eliminating HPV in the majority of cases, preventing the progression to cervical cancer in approximately 90% of infections."

Chesson HW, Dunne EF, Hariri S, Markowitz LE. The estimated lifetime probability of acquiring human papillomavirus in the United States. Sex Transm Dis. 2014 Nov;41(11):660-4. doi: 10.1097/OLQ.0000000000000193. PMID: 25299412; PMCID: PMC6745688.

https://pubmed.ncbi.nlm.nih.gov/25299412/

https://www.jostrust.org.uk/information/cervical-cancer/causes-risks

Line 383-384, "Unfortunately, the incidence of diseases caused by HPV infections is still alarming, leading CC to be the second most common cancer in women globally." => "Unfortunately, the high incidence of diseases resulting from HPV infections remains a major concern, contributing to cervical cancer (CC) being the fourth most common cancer among women worldwide."

https://www.who.int/news-room/fact-sheets/detail/cervical-cancer

Line 465-469, "Over 15 years since its introduction, the global acceptance of the HPV vaccine remains low, potentially linked to reported adverse reactions following vaccination. In Latin America, vaccination rates are below expectations, with several countries, including Brazil, Mexico, and Argentina, experiencing a significant decline in their vaccination rates. Moreover, adherence to the three-dose regimen in this region has notably been insufficient" => "Over 15 years since its introduction, the global uptake of the HPV vaccine has been lower than anticipated, which may be influenced by various factors including public perception of vaccine safety. In Latin America, the coverage of HPV vaccination is below expected levels, with countries such as Brazil, Mexico, and Argentina reporting significant declines in vaccination rates. This situation is further complicated by varying vaccination policies and access across the region. Additionally, the adherence to the previously recommended three-dose regimen has been challenging, though recent WHO recommendations now endorse a one or two-dose schedule for specific age groups, which could potentially improve vaccine accessibility and compliance."

https://www.who.int/news/item/20-12-2022-WHO-updates-recommendations-on-HPV-vaccination-schedule

Line 591-598, "A clinical study assessed the safety and immunogenicity of the AS04-adjuvanted HPV-16/18 vaccine in 4 to 6-year-old girls across Colombia, Mexico, and Panama over a 36-month period post-vaccination. The results indicated a sustained immunological response with detectable antibody levels against HPV-16 and HPV-18 in all girls, high antibody concentrations at 36 months, and no withdrawals due to adverse events. Due to the vaccine's well-tolerated nature and its ability to induce a sustained high immunological response, it has been proposed a pediatric HPV vaccination as a potential strategy to over come limitations in adolescent vaccination programs" => "A clinical study assessed the safety and immunogenicity of the AS04-adjuvanted HPV-16/18 vaccine in 4 to 6-year-old girls across Colombia, Mexico, and Panama over a 36-month period post-vaccination. The results indicated a sustained immunological response with detectable antibody levels against HPV-16 and HPV-18 in all girls, high antibody concentrations at 36 months, and no withdrawals due to adverse events. Due to the vaccine's well-tolerated nature and its ability to induce a sustained high immunological response, it has been proposed as a pediatric HPV vaccination strategy to overcome limitations in adolescent vaccination programs. HPV vaccine may be given at the same time as other vaccines. Concomitant administration of HPV vaccine with other vaccines in the children's vaccination program would minimize the number of visits required to deliver each vaccine individually."

Schilling A, Parra MM, Gutierrez M, Restrepo J, Ucros S, Herrera T, Engel E, Huicho L, Shew M, Maansson R, Caldwell N, Luxembourg A, Ter Meulen AS. Coadministration of a 9-Valent Human Papillomavirus Vaccine With Meningococcal and Tdap Vaccines. Pediatrics. 2015 Sep;136(3):e563-72. doi: 10.1542/peds.2014-4199. Epub 2015 Aug 3. PMID: 26240207.

https://pubmed.ncbi.nlm.nih.gov/26240207/

Line 732-780, "Discussion
Despite significant advancements in understanding Human Papillomavirus (HPV) and cervical cancer (CC), these remain formidable global health challenges. Extensive research has illuminated aspects of the infection, immune response, and pathogenesis, yet HPV continues to cause significant mortality worldwide. This underscores the critical need for enhanced prevention, treatment, and vaccination strategies.

Hormonal factors, notably estrogen and progesterone, are pivotal in the pathogenesis of CC, particularly in the context of prolonged oral contraceptive use. These hormones influence the integration of the viral genome into host cells, enhance the expression of HPV oncoproteins, and modulate interactions with hormonal receptors. Specifically, estrogen receptor (ER) signaling, especially ERα, plays a crucial role in the development of CC in both infected and stromal cells. The potential of ER antagonists and G protein-coupled estrogen receptors (GPER) as therapeutic targets is noteworthy, given their increased expression in CC and its precursors. Additionally, the role of prolactin (PRL) and its receptors in modulating key cellular functions such as migration and apoptosis in CC underscores the complex interplay of these hormonal pathways, presenting opportunities for targeted treatments.

Globally, the implementation of HPV vaccines has significantly reduced HPV-related cancers and infections. However, HPV continues to be a major health burden, and cervical cancer ranks high among cancers in women worldwide. Vaccines like Cervarix®, Gardasil, and Gardasil 9, targeting specific HPV serotypes, have effectively reduced cervical malignancies. Nevertheless, in regions like Latin America, HPV vaccination rates are suboptimal, hindered by factors such as parental concerns, cultural beliefs, and socioeconomic disparities.

To address these challenges, strategies including educational campaigns, enhanced training for healthcare providers, and community engagement initiatives have been proposed. Emphasizing family involvement in sexual health education and HPV awareness is crucial for improving vaccine acceptance and coverage.

Efforts to increase HPV vaccination coverage in Latin America, particularly among adolescents, involve collaborations and evidence-based policies. Successful models from countries like Colombia and Bolivia highlight the effectiveness of community engagement and data-driven approaches in increasing vaccine accessibility and acceptance.

In summary, while strides have been made in HPV vaccine development and implementation, continued research, strategic approaches, and effective execution are essential to achieve optimal vaccination coverage, overcome barriers, and ensure equitable access to HPV vaccines. This collective endeavor aims at the eventual eradication of HPV-related diseases.

Conclusion:
In conclusion, this manuscript emphasizes the ongoing global challenge posed by HPV and cervical cancer, highlighting the crucial role of hormonal factors in disease progression and the significant impact of HPV vaccines in reducing related cancers and infections. Despite advancements, challenges in vaccine coverage, particularly in Latin America, call for targeted strategies to improve awareness, acceptance, and accessibility of vaccinations. The manuscript advocates for continued research, strategic initiatives, and collaborative efforts to enhance vaccine coverage and ultimately aims towards the elimination of HPV-related diseases."

Comments on the Quality of English Language

Minor editing of English language required

Author Response

(The authors gave the same response as above.)

Reviewer 5 Report

Comments and Suggestions for Authors

-          Could the Authors the vaccine coverage in Latin American countries?

-          Line 84-86: It was just reported in the previous paragraph (Line 37-38). It can be removed.

-          The aim of the study is unclear. The Authors propose, about the role of HPV genes in viral replication, immune evasion, and impact in Latin American women, is too generic and not focused on a specific topic. Moreover, the Authors did not declare the methodology applied: is it a systematic or narrative review?

-          The 2nd paragraph did not report any news or add valuable information in the scenario of HPV-related disease and then, can be removed.  

-          I kindly suggest to the Authors to review all the manuscript starting with the definition of primary and secondary aims. If necessary, they could separate the macro area of the topics between epidemiological and immunological information in two articles, in order to be clearer for readers.

-          All review must be supported by a specific methodology, both in systematic and narrative review, but in this study the methodology is not included in the text.

-          The manuscript should be re-submitted after careful revision.

Comments on the Quality of English Language

Extensive editing of English language required

Author Response

(The authors gave the same response as above.)

Round 2

Reviewer 1 Report

Comments and Suggestions for Authors the authors have reconstructed their manuscript, making it more transparent even for a reader unfamiliar with the topic. I recommend this article for publication.
Good luck

Author Response

We are really thankful for your previous comments and we appreciate your approval for this article. Definitely, your suggestions made a better version of it.

Reviewer 2 Report

Comments and Suggestions for Authors

The paper has been greatly improved. I support accepting it for publication.

Comments on the Quality of English Language

None

Author Response

(The authors gave the same response as above.)

Reviewer 3 Report

Comments and Suggestions for Authors

Dear authors,

The text is more clear after editing. According to my opinion, the table 1 is still to abundant, but clearer than it was before. I recommend to reconsider the article after minor revision.

 Another comments

Lines 35-39

 HPV transmission occurs through skin-to-skin and skin-to-mucous membrane contact. Among the main routes are horizontal transmission (which includes fomites and skin, but not through sexual contact), autoinoculation, vertical transmission (from mother to newborn), and sexual transmission, the latter being the most documented and known [3].

Comment: Very good  written. You can add more: “The presence of HPV infection in one anatomical site represents the great risk for HPV infection in another anatomical site. The highest risk for cervical HPV infection poses the presence of anal HPV infection, and conversely, the presence of cervical HPV infection is a major risk factor for anal HPV infection [Lin 2019, Dzundova 2023].

 Lin et al. Cervical determinants of anal HPV infection and high-grade anal lesions in women: a collaborative pooled analysis. Lancet Infect Dis. 2019 Aug;19(8):880-891. doi: 10.1016/S1473-3099(19)30164-1. Epub 2019 Jun 13. PMID: 31204304; PMCID: PMC6656696.

 Dzundova et al. Risk Factors for the Anal and Oral Human Papillomavirus (HPV) Infections among Women with Severe Cervical Lesions: A Prospective Case-Control Study. Biomedicines. 2023 Nov 29;11(12):3183. doi: 10.3390/biomedicines11123183. PMID: 38137404; PMCID: PMC10741157.

Lines 148-150

Approximately 10% of women with an HPV infection develop a persistent infection, which can potentially lead to the formation of low- or high-grade intraepithelial lesions (Cervical Intraepithelial Neoplasia grade 2 or 3).

Comment: If you use one explanatory term, it is better to use the others as well. Nowadays, it is being advocated that CIN use LSIL and HSIL. The sentence should be changed as follows: “Approximately 10% of women with an HPV infection develop a persistent infection, which can potentially lead to the formation of low-grade intraepithelial lesions (LSIL or CIN 1, Cervical Intraepithelial Neoplasia grade 1) or high-grade intraepithelial lesions (HSIL or Cervical Intraepithelial Neoplasia grade 2 and 3).

Lines 219-221

Oral contraceptives are substances that fulfill the effect of natural hormones to prevent pregnancy by preventing the maturation of the ovarian follicle and thus preventing ovulation.

Comment:

Oral contraceptives are substances that do not fulfill the effect of natural hormones, on the contrary, they do not lead to maturation of the ovarian follicle. The sentence should be changed as follows: “Oral contraceptives prevent pregnancy by blocking the maturation of the ovarian follicle and thus preventing ovulation.“

Lines 330-334

Concerning HPV genotype distribution, the behavior is unpredictable in each country and even in the different regions into which a nation is divided; for example, a study carried out in three areas from Cuba (Holguin, Havana, and Villa Clara) found differences in the number of coinfections in which Holguin stands out for having a lower percentage of coinfections and a marked presence of HPV 89 compared to Havana and Villa Clara[86].

Comment: Better as follow: “Concerning HPV genotype distribution, the prevalence of particular genotypes is unpredictable in each country and even in the different regions into which a nation is divided; for example, a study carried out in three areas from Cuba (Holguin, Havana, and Villa Clara) found differences in the number of coinfections in which Holguin stands out for having a lower percentage of coinfections and a marked presence of HPV 89 compared to Havana and Villa Clara[86].

Table 1

Comment: Table 1 is significantly clearer, but still slightly confusing.

First: With the abbreviation HSIL, it is not clear whether these are patients with histologically proven HSIL of the cervix after biopsy (conization) or patients with cystology (PAP smear) HSIL. This should be clearly stated, eg HSIL (histology) or HSIL (cytology).

Second: If you want to have all papers on HPV distribution reported in some Latin American countries there, you should divide the tables into at least 3 sections: 1. only women 2. only men 3. mixed population

Lines 379-386

However, a Colombia study carried out in heterosexual couples in which the woman presented cervical intraepithelial lesions reported that only 28% of couples showed concordance in at least 1 HPV in contrast to 60% that did not give any HPV in common, which is probably due to differences between men and women in the resolve infection time. The prevalent high risk viral type in both men and women was HPV-16, with frequencies of 21.4% and 25%, respectively. On the other hand, the correlation of infection among partners was 40% for HPV positivity and 28% for type-specific HPV concordance.

Comment:  It does not make sense. See Abstract your citation Nr 94: “The prevalent high risk viral type in both men and women was HPV-16, with frequencies of 21.4% and 25%, respectively. On the other hand, the correlation of infection among partners was 40% for HPV positivity and 28% for type-specific HPV concordance.  This means that at least one of the two partners had no HPV detected in 60% of them. Moreover you repeat the same in one paragraph. You must rewrite the paragraph.

Author Response

We express our sincere gratitude for your valuable comments and insightful suggestions. After considering your comments and suggestions, we made proper modifications and responded to every point in the attached file. You can view the information inserted in the document highlighted in Yellow.
We would appreciate your feedback and, hopefully, your positive approval of this manuscript.

Reviewer 5 Report

Comments and Suggestions for Authors

Dear Authors,

I thank you for your answers. However, having clarified the purpose of the paper, I suggest removing paragraphs 1, 2, 3, and 4 because they are unrelated to the aims of the study and the setting of the review, which are women in Latin America. Although interesting, the information is generic and widely reported in the literature.

Moreover, in paragraph 5, the information regarding HPV prevalence in men should be removed. It could be discuss in the final paragraph, as discussion.

I strongly recommend to add a sentence about the benefit of self-collection to improve screening strategies. In particular, you can refer to recent study focused on feasability and acceptability of self-collection: doi: 10.3390/pathogens12091169.

I believe that these changes could notably improve the quality of the paper.

Best regards.

Comments on the Quality of English Language

Moderate english revision is required.

Author Response

We express our sincere gratitude for your valuable comments and insightful suggestions. After considering your comments and suggestions, we made proper modifications and responded to every point in the attached file. You can view the information inserted in the document highlighted in green.
We would appreciate your feedback and, hopefully, your positive approval of this manuscript.
